# Safety and efficacy of an iBTA-induced autologous Biotube vascular graft and its preparation device BTM1 in below-the-knee bypass surgery for chronic limb threatening ischemia: A protocol for an open-label, single-arm, multicenter clinical trial

Takashi Shuto[1]*, Ryuji Higashita[2], Hidenori Sako[3], Fumie Abe[4], Nozomi Yabuuchi[4], Tadashi Umeno[1], Kazuki Mori[1], Takayuki Kawashima[1], Yumiko Nakashima[1], Yasuhide Nakayama[5], Shinji Miyamoto[1]

**1** Department of Cardiovascular Surgery, Oita University Hospital, Yufu, Oita, Japan, **2** Department of Cardiovascular Surgery, Yokohama General Hospital, Yokohama, Kanagawa, Japan, **3** Department of Cardiovascular Surgery, Oita Oka Hospital, Oita, Oita, Japan, **4** Department of Development Promotion, Clinical Research, Innovation and Education Center (CRIETO), Tohoku University Hospital, Sendai, Miyagi, Japan, **5** Osaka Laboratory, Biotube Co., Ltd. 302 Kansai University Center for Innovation and Creativity, Suita, Osaka, Japan

* shutot@oita-u.ac.jp

## Abstract

### Purpose

Chronic limb-threatening ischemia (CLTI) increases the risk of lower limb amputation if revascularization is not performed. The use of autologous venous conduits is the only option for patients requiring below-the-knee bypass surgery, but it is limited by a lack of usable veins. The Biotube Maker (BTM1), based on in-Body Tissue Architecture (iBTA) technology, is a mold for the in vivo production of the Biotube® regenerative artificial vascular grafts. This clinical trial is designed to evaluate the safety and efficacy of subcutaneous embedding of the BTM1 for Biotube preparation and arterial bypass surgery using Biotube.

### Methods

Patients with CLTI who lack suitable veins for bypass surgery will be enrolled. This exploratory, investigator-initiated clinical trial will include 12 subjects. The primary endpoint is successful formation of an implantable Biotube following subcutaneous embedding of the BTM1. Secondary endpoints include intraoperative usability, patency and biocompatibility of the Biotube, wound healing, relief of rest pain, limb salvage, and procedure-related mortality, assessed up to 12 weeks after surgery. These outcomes are expected to provide essential feasibility and safety data to guide future pivotal studies.

**Data availability statement:** The minimal anonymized dataset underlying the findings of this study is available from the Zenodo repository at DOI: [10.5281/zenodo.17451392]. This article describes the study protocol; no clinical data have yet been generated.

**Funding:** Initials of the authors who received each award is SM. Grant numbers is C-65. The full name of funder is the Japan Agency for Medical Research and Development (AMED). The URL of funder website is "https://www.amed.go.jp/en/index.html". This is an investigator initiated clinical trial. Therefore, the funder does not have any role in the study design; collection, management, analysis, and interpretation of the data; writing of the manuscript; or the decision to submit the report for publication.

**Competing interests:** I have read the journal's policy and the authors of this manuscript have the following competing interests:YN (Yasuhide Nakayama) and RH (Ryuji Higashita) are employees and stockholders of Biotube Co., Ltd. The other authors declare that they have no competing interests. This does not alter our adherence to PLOS ONE policies on sharing data and materials.

## Discussion

This study may offer a new treatment option for CLTI patients who otherwise face major amputation. If feasibility and safety are confirmed, the findings will support planning of a pivotal trial aimed at regulatory approval.

## Trial registration

jRCT2072220062. Registered on October 19, 2022.

## SPIRIT schedule

Fig 1 shows the schedule of enrollment, interventions, and assessments in this study.

## Introduction

### Background and rationale

Chronic limb-threatening ischemia (CLTI) is characterized by rest pain, foot ulcers or gangrene, and increases the risk of lower-limb amputation if appropriate revascularization is not performed at an early stage. Revascularization of the lower limbs generally consists of either endovascular treatment, performed by catheter intervention, or bypass surgery using autologous veins or prosthetic grafts. Endovascular treatment of the tibial and pedal arteries below the popliteal level often results in restenosis and occlusion [1]. Bypass with small-diameter prosthetic graft has generally been associated with poor outcomes [2,3]. Although modifications such as the use of a Miller cuff have been attempted to improve patency, autologous venous conduits remain the gold standard, particularly for patients requiring below-the-knee bypass surgery [2,3]. However, their use is limited by the lack of suitable veins. Consequently, in the absence of proper veins, major amputation is often unavoidable. Amputation significantly worsens life prognosis [4]. In other words, the inability to perform bypass surgery directly threatens patient survival.

### BTM1 and biotube

In-body tissue architecture (iBTA), which was developed by Nakayama, is a cell-free, in vivo tissue engineering technology that can produce autologous implantable tissues of a desired shape using specially designed molds that are embedded subcutaneously [5,6]. BTM1 is an improved version of the 5th-generation Biotube Maker, which is a medical device used for the in vivo production of a regenerative vascular graft, called a Biotube, with a small diameter and long length for lower limb bypass surgery. A Biotube of 6 mm in diameter and 7 cm in length was reported to remain patent for 3 years in the first-in-human use in bypass surgery to dialysis shunt vessels [7]. The Maker was designated as a medical device in the SAKIGAKE program of the Japanese Ministry of Health, Labour and Welfare in 2019. The Maker has a porous disk shape of 86 mm or 97 mm diameter and 5.4 mm in thickness, which is assembled by sandwiching a spiral plastic rod between two stainless steel round plates [8,9]. The Maker was designed to form a Biotube of 40 cm or 55 cm in length with an inner diameter of 4 mm, with one end tapering to a diameter of 3 mm, and a thickness of 0.85 mm (Fig 2).

| | Entry | BTM1 Embedding | | BTM1 extraction | Biotube implantation | | | | | | |
|---|---|---|---|---|---|---|---|---|---|---|---|
| | | 0 day | 6 weeks | | 0 day | 1 day | 3 days | 1 week | 4 weeks | 8 weeks | 12 weeks |
| Inclusion/exclusion criteria | ✓ | | | | | | | | | | |
| Signed consent form | ✓ | | | | | | | | | | |
| Subject background | ✓ | | | | | | | | | | |
| Subjective and objective findings of the affected limb | ✓ | ✓ | ✓ | ✓ | ✓ | ✓ | ✓ | ✓ | ✓ | ✓ | ✓ |
| Blood test | ✓ | ✓ | ✓ | ✓ | ✓ | ✓ | ✓ | ✓ | ✓ | ✓ | ✓ |
| Lower limb artery echography | ✓ | | ✓ | | | | | ✓ | ✓ | ✓ | ✓ |
| Upper and lower limb vein echography | ✓ | | | | | | | | | | |
| Lower limb CTA* or DSA† | ✓ | | | | | | | ✓ | | | ✓ |
| ABI‡ | ✓ | | | | | | | ✓ | ✓ | ✓ | ✓ |
| SPP§ | ✓ | | | | | | | ✓ | ✓ | ✓ | ✓ |
| Evaluation of ischemic limbs | ✓ | | | | | | | ✓ | ✓ | ✓ | ✓ |
| Questionnaire | | | ✓ | | | | | | | | ✓ |
| Biotube sample storage | | | | ✓ | | | | | | | |
| Combination drug | ✓ | ✓ | ✓ | ✓ | ✓ | ✓ | ✓ | ✓ | ✓ | ✓ | ✓ |
| Adverse event | | ✓ | ✓ | ✓ | ✓ | ✓ | ✓ | ✓ | ✓ | ✓ | ✓ |

*CTAI: computed tomography angiography; †DSA: digital subtraction angiography; ‡ABI: ankle brachial pressure index; §SPP: skin perfusion pressure.

**Fig 1. SPIRIT schedule of enrollment, interventions, and assessments.**

## Non-clinical study

The following points were confirmed in the evaluation of the BTM1 in a non-clinical study using a goat model [8–10]. (1) Subcutaneous embedding was possible without damaging the BTM1 or the subcutaneous space. (2) The shape and strength required for Biotube formation were maintained, even when the BTM1 was implanted for several months. (3) A

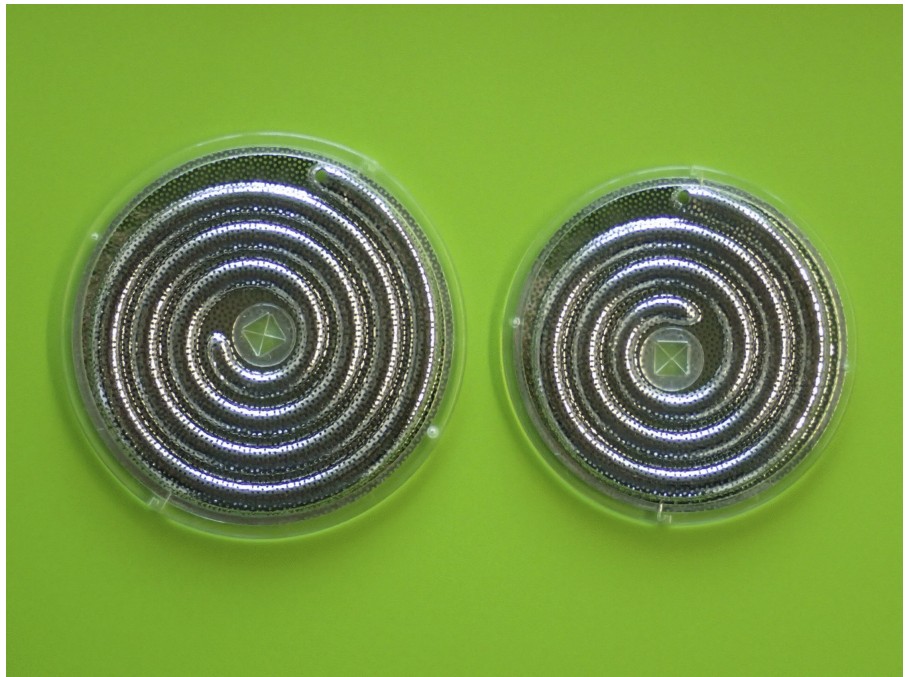

**Fig 2. BTM1.** The BTM1, composed of stainless steel and plastic components, is available in two sizes, yielding Biotubes with lengths of 40 cm (right side) and 55 cm (left side).

Biotube that could be used for surgical bypass surgery was formed in the BTM1 after at least 1 month of embedding. (4) The Biotube could be removed from the BTM1 without damage. (5) The biological safety of the BTM1 as an implantable medical device was demonstrated.

The formed Biotube is shown in Fig 3. Regarding the Biotube, the following points were also confirmed in the above-mentioned non-clinical study. (1) The Biotube had the shape and quality required for clinical application. (2) The Biotube had mechanical strength and flexibility. Thus, (3) the Biotube was implantable in goat carotid arteries. (4) After implantation, the vessel-constituting cells infiltrated the Biotube as a scaffold, and a three-layered vascular structure, including entire luminal endothelialization, was reconstructed within 3 months. (5) In the process of vascular regeneration, the balance between Bio-tube degradation and vascular wall reconstruction was maintained. (6) Long-term patency over 2 years was obtained without vascular deformation including aneurysm formation or anastomotic insufficiency. (7) The Biotubes could be connected to each other with little bleeding from the needle holes. (8) The Biotube was storable in an alcoholic solution for at least 6 months. In addition, case report on the first-in-human application of the Biotube in distal bypass for a CLTI patients was reported [11].

## Methods: Participants

### Objectives

This clinical trial is designed to evaluate the safety and efficacy of subcutaneous embedding of the BTM1 for Biotube preparation, and arterial bypass surgery at the lower limb for CLTI patients using Biotubes obtained from the BTM1.

### Trial design

This trial is a multicenter, open-label, uncontrolled, exploratory, medical device, investigator-initiated clinical study. There is no suitable alternative medical device for CLTI patients who require bypass surgery with saphenous veins. The purpose

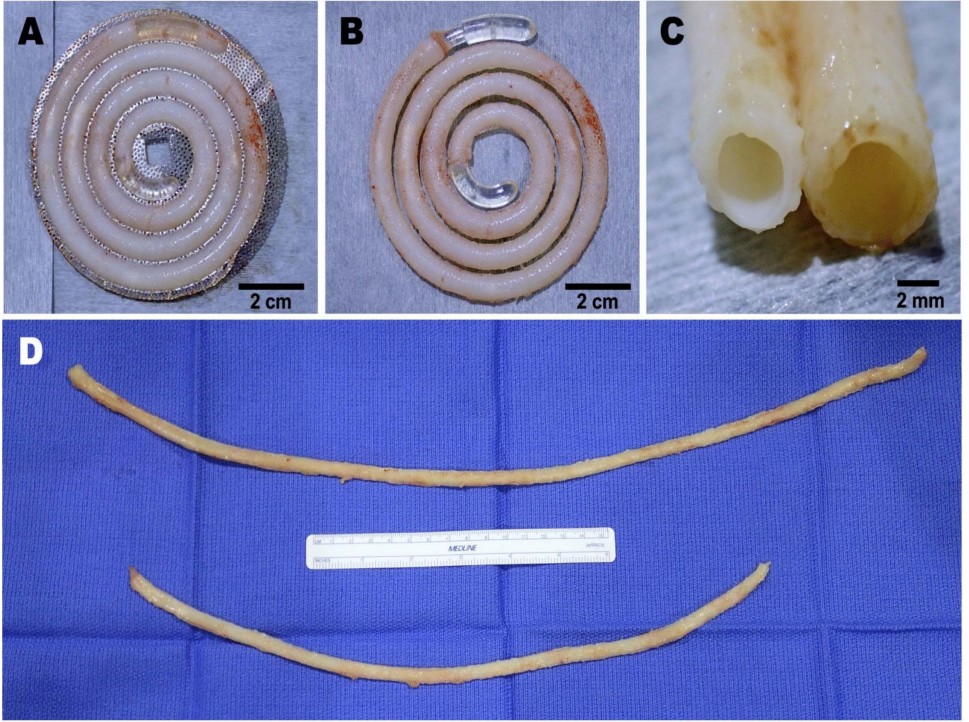

**Fig 3. Macroscopic view of Biotube obtained from BTM1. (A)** Biotube formed inside BTM1. **(B)** Biotube harvested from BTM1 with a spiral mandrel. **(C)** Both ends of Biotube with a diameter of 3 or 4 mm. **(D)** Whole view of a 55 cm (upper) or 40 cm (lower) long Biotubes obtained from two sizes of BTM1.

of this clinical study is to examine the efficacy and safety of a Biotube prepared subcutaneously in human subjects using BTM1 for bypass surgery to the lower leg artery. Therefore, it was decided to conduct the study under an open-label, uncontrolled design.

## Study setting

The planned study was submitted to the Pharmaceuticals and Medical Devices Agency (PMDA), the Japanese regulatory agency, for August 2022. This trial has already been approved by the Institutional Review Boards at each study facility (Oita University Hospital, Yokohama General Hospital, and Oita Oka Hospital in Japan). The clinical trial participation period for each subject shall be from the date of obtaining consent for participation to the end of the observation period (12 weeks after Biotube implantation). The observation period was determined based on the fact that bypass surgery using the saphenous vein usually improves symptoms in 12 weeks. After the observation period in this study, a 5-year follow-up will be conducted as clinical research.

## Trial status

The trial was registered with the Japan Registry of Clinical Trials (jRCT) on October 19, 2022, as protocol jRCT2072220062. The protocol is version 2, created on September 5, 2022. Study procedures were approved by the Institutional Review Board (IRB) of Oita University Hospital (A22-002) on July 27, 2022; Yokohama General Hospital (C22-001) on July 28, 2022; and Oita Oka Hospital (C22-002) on July 29, 2022.

### Timeline of the study

Recruitment of subjects began on September 20, 2022. The end date for subject recruitment is scheduled for November 30, 2025. Data collection is expected to be completed by May 2026. Results from the study are expected to be reported by December 2026.

### Sample size

The planned number of subjects is 12. There are two types of test devices (large and small), and the implant location can be selected from four possible sites: the chest, abdomen, buttocks, or thighs. Because the implant site and number of implants are determined by the investigators based on the condition judged most suitable for each subject, it is not feasible to equally distribute cases across all possible device–site combinations. This study is designed as an exploratory trial to evaluate feasibility and safety rather than to statistically compare all possible combinations. In order to ensure that each device type and implantation site can be examined in more than one subject under real-world clinical decision-making, we considered a minimum of 12 subjects to be necessary. Taking into account feasibility and the potential for interim analysis, we therefore set a target enrollment of 12 subjects.

### Explanation for the choice of comparators

There is no suitable alternative medical device for CLTI patients who require bypass surgery with saphenous veins. The fate of the affected limb if revascularization is not adequately performed is also clear. Therefore, we decided that a control group was ethically unnecessary.

### Eligibility criteria

The complete inclusion and exclusion criteria are shown in Table 1. Candidate patients who meet the inclusion and exclusion criteria for enrolment are registered after obtaining their informed consent. In addition to the principal investigator, it is essential for enrolment to consult third parties with appropriate expertise (e.g., endovascular interventionists, wound management specialists, etc.) for registration.

### Informed consent

The principal investigators will provide the subjects with sufficient information prior to their participation in this study using an IRB-approved explanatory document. The principal investigators will then obtain written consent from the subject to participate in this study. Prior to offering their consent, subjects will be given the opportunity to ask questions and the time necessary to decide whether or not to participate in this study. Participation in this clinical trial is voluntary.

## Interventions and outcomes

### Intervention description

Surgery is performed under general anesthesia. An incision is made in the skin of the chest, abdomen, buttocks or thighs to create a subcutaneous pocket beneath the dermis layer. The minimum number of BTM1s to be implanted is 2, and the maximum number is 4. It is essential to implant one in the abdomen. The size of BTM1 is selected at the discretion of the investigator according to the physical constitution of the subject. A sizar is inserted into the created subcutaneous pocket, to confirm that there is sufficient space for BTM1 implantation. The BTM1 is then inserted into the subcutaneous pocket with connecting a Blake® silicon drain (Johnson & Johnson, Inc. New Brunswick) before closing the wound. The air in the BTM1 is sucked from the Blake® silicon drain and the BTM1 is brought into close contact with the tissue. At 2–7 days after implantation, an echo examination is performed to confirm that there is no subcutaneous fluid, and the Blake® silicon drain is removed. The BTM1 is indwelled for 4–24 weeks (≥8 weeks in principle) after implantation. Under general anesthesia,

**Table 1. Inclusion and exclusion criteria.**

| Inclusion criteria | |
|---|---|
| 1 | Agreement for participating in the study and informed consent signed by the patient. |
| 2 | Age is over 18 years old. |
| 3 | Lower limb ischemia that meets grade 2 or 3 (ABI*<0.6, SPP†<40 mmHg) in Wifi classification of CLTI‡. |
| 4 | Patients correspond to one of the following. |
| | a) GLASS§ classification of GVG‖ is stage III, and corresponds to clinical stage 2, 3 or 4 of Wifi classification. |
| | b) GLASS classification is stage II, and corresponds to clinical stage 3 or 4 of Wifi classification. |
| | c) GLASS classification is stage I or II, and clinical symptoms do not improve even if intravascular treatment is performed. |
| 5 | Upper or lower limb veins of the optimal length and diameter required for bypass surgery does not exist. |
| 6 | Patients with arterial occlusion on the proximal side of the planned proximal anastomosis site for bypass surgery. |
| 7 | Survival of 12 months or more and followed-up for 12 weeks after bypass surgery are possible. |
| Exclusion criteria | |
| 1 | Patients who have difficulty in securing the implantation period of the clinical trial equipment required for Biotube formation. |
| 2 | Patients with general condition who are difficult to tolerate surgery due to severe malnutrition and complications. |
| 3 | Patients with poor skin condition who cannot secure more than one implantation site for the investigational device. |
| 4 | Patients undergoing invasive surgery within 30 days prior to enrollment |
| 5 | Patients who do not have a peripheral target artery that can be bypassed or who have undergone endovascular treatment at the planned anastomosis site on the peripheral side. |
| 6 | Patients with arterial occlusion on the proximal side of the planned proximal anastomosis site for bypass surgery. |
| 7 | Patients who cannot confirm blood flow on the distal side of the planned peripheral anastomosis site for bypass surgery. |
| 8 | Patients with lower limb amputation proximal to the metatarsals. |
| 9 | History and complications of malignant tumors (excludes those with no recurrence for more than 5 years after treatment or new onset). |
| 10 | Patients who use immunosuppressants for autoimmune diseases and post-implantation. |
| 11 | Patients who have a history of allergies to stainless steel or polyolefin resin. |
| 12 | Pregnancy. |
| 13 | Participating in other clinical trials. |
| 14 | Patients judged by the investigator to be inappropriate due to medical conditions or safety reasons. |

*ABI: ankle brachial pressure index; †SPP: skin perfusion pressure; ‡CLTI: chronic limb threatening ischemia; §GLASS: Global Anatomic Staging System; ‖GVG: Global Vascular Guideline.

the skin is incised and the BTM1 is removed. After removing the connective tissue around the outer circumference of the BTM1, the outer shell parts are removed, and the Biotube formed inside BTM1 is taken out together with the core part. The Biotube is then withdrawn from the core part and washed with a saline solution, followed by immersion in a dish filled with a 70% alcohol solution for 30 min after inserting a straightening rod into its lumen. The Biotube is then washed again with a saline solution and visually inspected to ensure there are no tears, defects, or severe unevenness in the wall thickness.

The Biotube is confirmed to have no leakage by injecting physiological saline into its lumen with a syringe, and the pressure resistance at 200 mmHg is confirmed using a pressure monitor. Furthermore, the ultimate strength is measured using a tensile tester (Stency; AcroEdge, Osaka, Japan) to confirm that the Biotube meets the standard strength (5N). The prepared Biotube is implanted into the lower limbs of the subject as an alternative blood vessel in the same manner as an autologous vein graft. Heparin is administered for 7 days after Biotube implantation to prevent the formation of blood clots. Heparin is then switched to oral anticoagulants and oral anticoagulant therapy continues until 12 weeks after implantation.

If removal of the study device and Biotube implantation cannot be performed on the same day for reasons on the part of the subject, temporary storage of the obtained Biotube is permitted. For temporary storage, the Biotube is kept sterile, soaked in a 10% alcohol solution, sealed and stored at room temperature. In this case, the temporary storage period shall be a maximum of 4 weeks, and Biotube implantation shall be performed within 4 weeks after removal of the investigational device.

Subjects will be followed up according to the schedule for up to 12 weeks after implantation of the Biotube. A summary of the study design is shown in Fig 4.

## Criteria for discontinuing or modifying allocated interventions

The study will be discontinued if any of the following apply. [1] If an adverse event occurs and the coordinating investigator or principal investigator determines that it is difficult to continue the study, or if the patients' original disease worsens; [2] No Biotube is formed for bypass surgery; [3] The subject withdraws their consent to participate in the study; [4] The subject is found to be ineligible after enrollment; [5] Significant protocol deviations are found; or [6] There is no need for bypass surgery 24 weeks after BTM1 embedding.

## Provisions for post-trial care

Medical expenses during the clinical trial period will be covered by the medical insurance of the subjects, excluding the test device and treatment/examination expenses that the coordinating investigator has agreed will be borne by the medical institution. Should a subject suffer health damage due to the implementation of this clinical trial, the medical institution will take necessary measures, such as the administration of appropriate treatment. In the event of such health damage, the investigator will provide appropriate compensation.

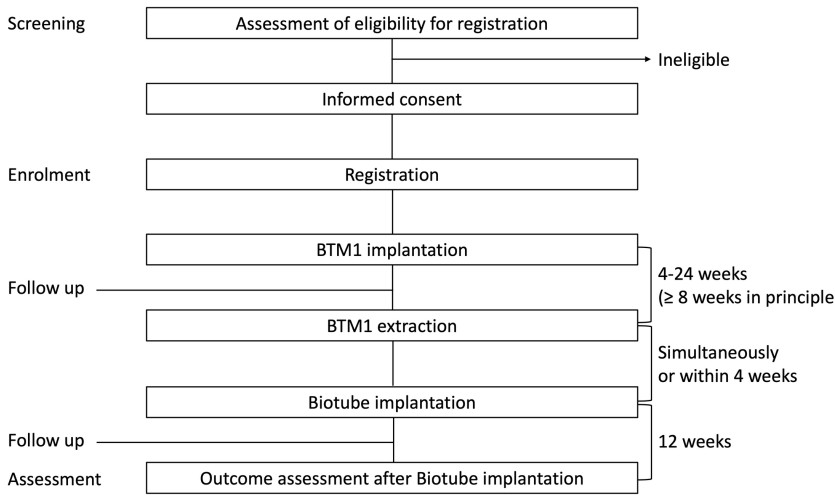

**Fig 4. Summary of the study design.**

## Outcomes

The primary efficacy endpoint is the successful formation of an implantable Biotube, defined as retrieval of a structurally intact Biotube of sufficient length and quality judged suitable for use in bypass surgery by the investigators.

The Secondary efficacy endpoints are:

(1) Successful subcutaneous embedding and retrieval of BTM1

(2) Technical success of bypass surgery using the obtained Biotubes

(3) Primary graft patency at 12 weeks after implantation

(4) Improvement of lower limb symptoms (rest pain, ulcer healing)

(5) Freedom from major amputation of the target limb

(6) Incidence of additional treatments required during BTM1 embedding and after Biotube implantation

(7) Change in quality of life scores during BTM1 embedding and after Biotube implantation

(8) Graft diameter after bypass

The safety endpoints are:

(1) Incidence of inflammation, oncogenesis, or mortality during BTM1 embedding

(2) Biocompatibility of implanted Biotubes

(3) Biotube rupture

(4) Death within 12 weeks after Biotube implantation

(5) Other adverse events

## Data collection and management

### Plans for assessment and collection of outcomes

The schedule of enrollment, interventions, and assessments is shown in Fig 1. The test data required at each timing is collected by the investigator. The quality of the Biotubes is evaluated by their appearance, pressure resistance, and strength to determine their suitability for implantation. Biotubes of approximately 0.5 to 2 cm in length are collected for the strength evaluation or histological observation. If the clinical trial is halted prior to implantation therapy, the appearance, strength, and histopathology of Biotubes obtained from the removed BTM1 are to be evaluated. All data acquired during the study will be anonymized and saved in a study folder on our protected research server. Only the study team has access to this specific study folder.

### Plans to promote participant retention and complete follow-up

The patients receive extensive information about the study set-up and requirements prior to study enrollment. The importance of the completion of follow-up is stressed. Patients are allowed to withdraw at any time during the study and are not obliged to give a reason for their desire to discontinue the study. Post-bypass follow-up is performed on an outpatient basis. The follow-up is completed while contacting the patients. If the clinical trial is discontinued due to an adverse event, in principle, patients are followed up until recovery or remission.

## Data management

Patient data are stored in study folders on each facility's secure study server, and backups are made on a regular basis (once every three months). Signed study consent forms are kept in a locked room at each facility. Paper-based media, such as questionnaires, are also kept in locked rooms at each facility. The entered data are double-checked by the coordinating investigator and the principal investigators. The clinical trial medical institution must retain the clinical trial documents and source materials to be preserved at the medical institution for three years after the completion of the clinical trial or until the date of manufacturing and marketing approval for the test device. The IRB establisher must keep records such as operating procedures, a list of committee members, a list of committee members' occupations and affiliations, submitted documents, meeting records, their summaries, and letters for three years after the completion of the clinical trial or until the date of manufacturing and marketing approval for the test device.

## Confidentiality

Research data are stored using a study identification code for each participant. The medical institution conducting the clinical trial creates a correspondence table between the personal identification information and the participant identification code. This correspondence table is properly managed by each medical institution. No identifying patient details are to be reported in any publication.

# Statistical methods

## Statistical methods for primary and secondary outcomes

A detailed statistical analysis plan will be prepared before database lock, and analyses will be conducted after database lock.

For the primary endpoint, the success rate will be calculated as the proportion of subjects from whom a Biotube meeting the predefined criteria for implantation (structurally intact, sufficient length, and judged suitable by the investigators) can be retrieved. This proportion will be presented together with the corresponding 95% confidence interval.

For the secondary efficacy endpoints, descriptive statistics will be provided. Specifically, technical success rates of bypass surgery, primary graft patency rates at 12 weeks, limb salvage rates, and changes in quality of life will be summarized using frequencies, percentages, means, and standard deviations, as appropriate. The incidence of additional treatments and graft diameters will also be analyzed descriptively.

For the safety endpoints, the incidence of adverse events, device malfunctions, Biotube rupture, and mortality will be tabulated and summarized descriptively.

Given the exploratory nature of this trial, no formal hypothesis testing is planned. All analyses will be descriptive in nature, and results will be interpreted accordingly.

## Interim analyses

An interim analysis will be performed when 6 subjects have reached 12 weeks after Biotube implantation. The primary purpose of this interim analysis is safety monitoring and feasibility assessment, rather than hypothesis testing or futility analysis. Specifically, the ability of the investigational device to form Biotubes and the patency of the Biotubes at 12 weeks will be descriptively summarized to confirm that the procedure is technically feasible and does not raise unexpected safety concerns. Given the exploratory nature of this study and the limited sample size, no formal statistical testing will be performed in the interim analysis, and the results will be used solely to guide the continued safe conduct of the trial.

# Oversight and monitoring

## Composition of the coordinating centre and trial steering committee

This trial is a multicenter study conducted at three facilities. Day to day support for the trial is provided as follows: Principal investigators are assigned to each facility, where they supervise the trial and assume medical

responsibility for the patients. The data manager organizes and protects the data obtained. The coordinating investigator is responsible for study registration, intersite coordination, and annual safety reporting. The principal investigators are also obtained informed consent from the participants and ensure follow-up is performed according to protocol. The research team meets every three months. There is no trial steering committee or stakeholder or public involvement group.

## Composition of the data monitoring committee, its role and reporting structure

The data and safety monitoring committee, which consists of third-party members (vascular surgeons of Asahikawa medical university, Kobe University and Yamagata prefectural central hospital in Japan), reviews each patient and decides whether or not to continue the clinical trial one month after Biotube implantation. This committee is independent from the sponsor and competing interests.

## Adverse event reporting and harms

All adverse events reported by the subjects or observed by the investigators are recorded. The causal relationship with the study treatment event is also recorded. The adverse event collection period ranges from the point of treatment with the study device until the end of the observation period or discontinuation by each individual subject.

Defects refer to problems with the specifications of the investigational device, defects or failures of the investigational device, problems with the procedure, etc. Information regarding malfunctions that occur during the clinical trial period are collected as needed. Based on the characteristics of this clinical trial, we continue to investigate defects in the Biotube even after removal of the investigational device.

In the event of a serious adverse event or a malfunction that may lead to a serious adverse event, the principal investigators report it to the head of the clinical trial site and the coordinating investigator. If the coordinating investigator determines that the adverse event is subject to reporting, as stipulated in Article 273 of the Enforcement Regulations of the Pharmaceuticals and Medical Devices Act, the coordinating investigator convenes the data and safety monitoring committee and considers whether or not to continue the study. In addition, if it is determined that the defect should be reported to the Minister of Health, Labor and Welfare, it is reported to the PMDA.

## Frequency and plans for auditing trial conduct

Audits are to be conducted by auditors of Accerise Corporation (Tokyo, Japan) at three points: when preparations for the clinical trial are completed, immediately after the first case is performed at each medical institution, and at the end of the clinical trial.

Audits performed when clinical trial preparations are completed mainly include inspection of documents and interviews with relevant parties. The audit when the first clinical trial is conducted involves visiting each medical institution. The system that conducts the clinical trial and the clinical trial materials and records are also audited. The inspection at the end of the clinical trial mainly involves inspection of documents and interviews with relevant parties. Items related to case report forms and records related to the implementation of monitoring are also audited.

## Plans for communicating important protocol amendments to relevant parties

If important new information becomes available that may be relevant to the subject's willingness to consent, the investigator promptly revises the explanatory document based on that information and obtains IRB approval. The coordinating investigator or principal investigators explain the details of the study to each patient once again using the revised explanatory document. Written consent is then obtained from the subject to confirm their continued participation in the study.

## Dissemination plans

The main results of this study will be submitted for publication in international peer-reviewed journals. Both positive and negative results will be reported.

## Discussion

This clinical trial is designed to evaluate the safety and efficacy of subcutaneous embedding of the BTM1 for Biotube preparation, and arterial bypass surgery at the lower limb for CLTI patients using Biotubes obtained from the BTM1. These findings may be promising for patients with CLTI who have no choice except to have their legs amputated. This study should be followed by a pivotal study to obtain satisfactory data for an application for marketing approval.

### Strengths and limitations

1) Biotube Maker (BTM1), which is based on in-Body Tissue Architecture technology, is a mold for production of the Biotube.

2) Biotube is a small diameter, long regenerative vascular graft that can be used for lower limb bypass in patients with critical limb ischemia.

3) A major strength of this study is evaluating the efficacy and safety of new technology-based "BTM1" and "Biotube".

4) The main limitations are the small number of patients and the lack of a control group.

## Supporting information

**S1 File. This is the SPIRIT checklist.**
(DOCX)

**S2 File. This is the original protocol of Biotube.**
(PDF)

**S3 File. This is the translated protocol of Biotube.**
(PDF)

## Author contributions

**Conceptualization:** Yasuhide Nakayama, Shinji Miyamoto.

**Data curation:** Ryuji Higashita, Hidenori Sako, Tadashi Umeno, Kazuki Mori, Takayuki Kawashima.

**Funding acquisition:** Yasuhide Nakayama, Shinji Miyamoto.

**Investigation:** Takashi Shuto, Ryuji Higashita, Hidenori Sako, Tadashi Umeno, Kazuki Mori, Takayuki Kawashima.

**Methodology:** Fumie Abe, Nozomi Yabuuchi, Yasuhide Nakayama, Shinji Miyamoto.

**Project administration:** Yumiko Nakashima.

**Resources:** Yasuhide Nakayama.

**Supervision:** Shinji Miyamoto.

**Validation:** Ryuji Higashita, Hidenori Sako, Yasuhide Nakayama.

**Visualization:** Yasuhide Nakayama.

**Writing – original draft:** Takashi Shuto.

**Writing – review & editing:** Takashi Shuto.

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
