## [Decision Letter · Decision Letter 0]

4 Sep 2025

Dear Dr. Shuto,

Thank you for submitting your manuscript to PLOS ONE. After careful consideration, we feel that it has merit but does not fully meet PLOS ONE’s publication criteria as it currently stands. Therefore, we invite you to submit a revised version of the manuscript that addresses the points raised during the review process.

2 reviewers evaluated your paper, so please revise it.

We look forward to receiving your revised manuscript.

Kind regards,

Yoshiaki Taniyama, MD, PhD

Academic Editor

PLOS ONE

Journal Requirements:

2. Thank you for stating the following in the Competing Interests/Financial Disclosure section:

I have read the journal's policy and the authors of this manuscript have the following competing interests: Yasuhide Nakayama and Ryuji Higashita are employees and stockholders of Biotube Co., Ltd..

The other authors have declared that no competing interests exist.

We note that one or more of the authors are employed by a commercial company: Biotube Co., Ltd

Reviewers' comments:

Reviewer's Responses to Questions

**Comments to the Author**

1. Does the manuscript provide a valid rationale for the proposed study, with clearly identified and justified research questions?

Reviewer #1: Yes

Reviewer #2: Yes

Reviewer #3: Yes

2. Is the protocol technically sound and planned in a manner that will lead to a meaningful outcome and allow testing the stated hypotheses?

Reviewer #1: Yes

Reviewer #2: Yes

Reviewer #3: Yes

3. Is the methodology feasible and described in sufficient detail to allow the work to be replicable?

Reviewer #1: Yes

Reviewer #2: Yes

Reviewer #3: Yes

4. Have the authors described where all data underlying the findings will be made available when the study is complete?

Reviewer #1: No

Reviewer #2: Yes

Reviewer #3: Yes

5. Is the manuscript presented in an intelligible fashion and written in standard English?

Reviewer #1: Yes

Reviewer #2: Yes

Reviewer #3: Yes

You may also provide optional suggestions and comments to authors that they might find helpful in planning their study.

Reviewer #1: I cannot accept this article because it didn't present any results; it only described the protocol of the clinical trial.

Reviewer #2: This work is very interesting because finding substitutes for the absent saphenous vein is essential for surgery on patients with critical ischemia.

I have only few suggestions.

Abstract: In the methods section, I would like to highlight some key findings from the study.

Methods: The problem is the time it takes for the tube to form; patients with severe critical ischemia cannot always wait three months.

What postoperative therapy should be used?

What would have been the alternative technique for the patients who will be enrolled?

With what other biomaterials will you compare the results obtained?

Reviewer #3: Page 4 lines 97-100. Please revise the description since prosthetic graft with Miller cuff may be also performed.

Page 4 line 100. Amputation significantly worsens the life prognosis

Please include references.

Table 1

Inclusion criteria 3. Ischemia grades range from 0 to 3, and WIfI is a stage, not a grade.

Page 9 lines 230-231. Will the graft harvesting and bypass be performed on the same day? What circumstances would be expected if the same-day implementation is not possible?

**Do you want your identity to be public for this peer review?** For information about this choice, including consent withdrawal, please see our Privacy Policy

Reviewer #1: No

Reviewer #2: **Yes: ** Francesco Stilo

Reviewer #3: **Yes: ** Koichi Morisaki

---

## [Author Response · Author response to Decision Letter 1]

12 Sep 2025

Reviewer #1:

Comment: I cannot accept this article because it didn't present any results; it only described the protocol of the clinical trial.

Response: We thank the reviewer for this comment. We respectfully note that this manuscript is submitted as a study protocol, in accordance with the journal’s scope and guidelines. As such, it is intended to describe the study rationale, objectives, and methodology in detail prior to the completion of the trial, and it does not include study results at this stage. We believe that the protocol format is appropriate and valuable for ensuring transparency, reproducibility, and scientific rigor in the planned clinical trial.

Reviewer #2:

Comment 1: Abstract: In the methods section, I would like to highlight some key findings from the study.

Response: Thank you for your helpful comment. We have revised the Abstract accordingly. In the Methods section, we now specify the primary endpoint (successful formation of an implantable Biotube following subcutaneous embedding of the BTM1) and the key secondary endpoints (intraoperative usability, patency and biocompatibility of the Biotube, wound healing, relief of rest pain, limb salvage, and procedure-related mortality). These revisions clarify the planned outcomes to be assessed in this exploratory trial, in line with your suggestion.

Comment 2: Methods: The problem is the time it takes for the tube to form; patients with severe critical ischemia cannot always wait three months. What postoperative therapy should be used?

Response: We thank the reviewer for this important comment. We fully agree that patients with severe critical ischemia who require immediate revascularization cannot wait for the three-month subcutaneous embedding period necessary for Biotube formation. To address this, our exclusion criteria clearly state that patients “in whom it is difficult to ensure the implantation period required for Biotube formation due to systemic conditions, such as the need for urgent revascularization,” are excluded from the study.

With regard to postoperative management, appropriate antithrombotic therapy is provided to prevent thrombosis of the implanted Biotube. Specifically, heparin is administered for 7 days after Biotube implantation, followed by a switch to oral anticoagulants, which are continued until 12 weeks after implantation. This therapeutic regimen is described in the Intervention section of the protocol.

We believe these criteria and postoperative management adequately address the reviewer’s concerns regarding patient selection and therapy after Biotube implantation.

Comment 3: What would have been the alternative technique for the patients who will be enrolled?

Response: We thank the reviewer for this important comment. The patients to be enrolled in this study are those with chronic limb-threatening ischemia (CLTI) who require distal bypass surgery to tibial or pedal arteries but have no suitable autologous venous conduit. For such patients, there is currently no established pharmacological therapy with proven efficacy, and no suitable alternative medical device is available. If revascularization cannot be adequately performed, the only remaining option is major limb amputation.

As stated in the “Explanation for the choice of comparators” section, “There is no suitable alternative medical device for CLTI patients who require bypass surgery with saphenous veins. The fate of the affected limb if revascularization is not adequately performed is also clear.” Therefore, patients eligible for this trial essentially have no alternative to bypass surgery with a Biotube graft, aside from amputation.

Comment 4: With what other biomaterials will you compare the results obtained?

Response: We thank the reviewer for this question. In this trial, no comparator biomaterial will be used. The efficacy of autologous vein grafts for below-the-knee bypass is already well established in the literature, and patients enrolled in this study are those who lack suitable autologous veins. Therefore, there is no alternative biomaterial available for comparison, and the present study is designed as a single-arm exploratory trial to evaluate the feasibility, safety, and efficacy of the Biotube graft.

Reviewer #3:

Comment 1: Page 4 lines 97-100. Please revise the description since prosthetic graft with Miller cuff may be also performed.

Response: We thank the reviewer for this helpful comment. In the revised manuscript, we have clarified that although small-diameter prosthetic grafts are generally associated with poor outcomes, modifications such as the Miller cuff have been attempted. However, autologous venous conduits remain the gold standard for below-the-knee bypass surgery, and their use is limited by the availability of suitable veins. The revised text now reads as follows:

“Bypass with small-diameter prosthetic grafts has generally been associated with poor outcomes. Although modifications such as the use of a Miller cuff have been attempted to improve patency, autologous venous conduits remain the gold standard, particularly for patients requiring below-the-knee bypass surgery. However, their use is limited by the lack of suitable veins. Consequently, in the absence of proper veins, major amputation is often unavoidable. Amputation significantly worsens life prognosis. In other words, the inability to perform bypass surgery directly threatens patient survival.”

Comment 2: Page 4 line 100. Amputation significantly worsens the life prognosis. Please include references.

Response: We thank the reviewer for this helpful suggestion. In accordance with the comment, we have added the following reference to support the statement that amputation significantly worsens life prognosis:

Leinweber ME, Greistorfer E, Rettig J, Taher F, Kliewer M, Assadian A et al. Quantification of the survival disadvantage associated with major amputation in patients with peripheral arterial disease. J Clin Med. 2025;14:104.

This citation has been incorporated into the revised manuscript.

Comment 3: Table 1, Inclusion criteria 3. Ischemia grades range from 0 to 3, and WIfI is a stage, not a grade.

Response: We thank the reviewer for pointing this out. You are correct that in the SVS WIfI classification, each component (Wound, Ischemia, and foot Infection) is graded from 0 to 3, and there is no grade 4. In our original text, we mistakenly wrote “grade 3 or 4” for ischemia. We have revised this to correctly read “Lower limb ischemia that meets grade 2 or 3 (ABI < 0.6, SPP < 40 mmHg) in WIfI classification of CLTI”. We appreciate the reviewer’s careful observation, which has improved the accuracy of our description.

Comment 4: Page 9 lines 230-231. Will the graft harvesting and bypass be performed on the same day? What circumstances would be expected if the same-day implementation is not possible?

Response: We thank the reviewer for this important question. In principle, removal of the Biotube and bypass surgery are performed on the same day under general anesthesia. However, if the patient’s general condition is poor and prolonged anesthesia would present an excessive burden, the bypass procedure may be performed on a different day. In such cases, implantation of the Biotube must be carried out within 4 weeks of its removal.

As described in the “Intervention description” section, if removal of the investigational device and Biotube implantation cannot be performed on the same day for reasons on the part of the subject, temporary storage of the Biotube is permitted. The Biotube is kept sterile, soaked in a 10% alcohol solution, sealed, and stored at room temperature. The maximum storage period is 4 weeks, and Biotube implantation must be performed within this period.

We believe this approach ensures both patient safety and the feasibility of the procedure in cases where same-day surgery is not possible.

---

## [Decision Letter · Decision Letter 1]

24 Sep 2025

Dear Dr. Shuto,

Thank you for submitting your manuscript to PLOS ONE. After careful consideration, we feel that it has merit but does not fully meet PLOS ONE’s publication criteria as it currently stands. Therefore, we invite you to submit a revised version of the manuscript that addresses the points raised during the review process.

We look forward to receiving your revised manuscript.

Kind regards,

Yoshiaki Taniyama, MD, PhD

Academic Editor

PLOS ONE

Journal Requirements:

Reviewers' comments:

Reviewer's Responses to Questions

**Comments to the Author**

1. Does the manuscript provide a valid rationale for the proposed study, with clearly identified and justified research questions?

Reviewer #2: Yes

Reviewer #3: Yes

Reviewer #4: Partly

2. Is the protocol technically sound and planned in a manner that will lead to a meaningful outcome and allow testing the stated hypotheses?

Reviewer #2: Yes

Reviewer #3: Yes

Reviewer #4: Partly

3. Is the methodology feasible and described in sufficient detail to allow the work to be replicable?

Reviewer #2: Yes

Reviewer #3: Yes

Reviewer #4: Yes

4. Have the authors described where all data underlying the findings will be made available when the study is complete?

Reviewer #2: Yes

Reviewer #3: Yes

Reviewer #4: Yes

5. Is the manuscript presented in an intelligible fashion and written in standard English?

Reviewer #2: Yes

Reviewer #3: Yes

Reviewer #4: Yes

You may also provide optional suggestions and comments to authors that they might find helpful in planning their study.

Reviewer #2: now it sounds good

all my questions has been answered

your paper has been well presented, and methods are very clear

Reviewer #3: I think that The revision appears to have improved the paper. Thank you for responding to my comments.

Reviewer #4: The justification of the sample size is unclear. With two device types and four implant locations, there are up to eight possible combinations. With only 12 subjects, these combinations will not be fully represented.

Outcomes: The definition of the primary endpoint is too generic and not consistent with its statistical definition.

The protocol should clearly specify the statistical method planned for analyzing the primary endpoint. If no hypothesis testing is not planned for the exploratory study, it needs to state that clearly. Descriptive measures such as success rates can be reported.

The purpose of the interim analysis is also unclear. Is it intended for safety monitoring or a futility check? With only half of the planned sample size, the analysis may not have sufficient power to draw meaningful conclusions.

**Do you want your identity to be public for this peer review?** For information about this choice, including consent withdrawal, please see our Privacy Policy

Reviewer #2: No

Reviewer #3: **Yes: ** Koichi Morisaki

Reviewer #4: No

---

## [Author Response · Author response to Decision Letter 2]

26 Sep 2025

Comment 6: Review Comments to the Author

Reviewer #4:

Comment 1: The justification of the sample size is unclear. With two device types and four implant locations, there are up to eight possible combinations. With only 12 subjects, these combinations will not be fully represented.

Response: We thank the reviewer for this important comment regarding the justification of the sample size. We agree that there are two device types and four possible implantation sites, resulting in up to eight potential combinations. However, because the choice of device size and implantation site is determined by the investigators according to the condition most suitable for each subject, it is not feasible to equally distribute the cases across all combinations in this exploratory study. The purpose of this trial is not to statistically compare all possible combinations, but rather to evaluate feasibility and safety under practical clinical conditions. In order to allow for the assessment of each condition (device type and implantation site) in more than one subject and to maintain feasibility of trial conduct, we determined that a minimum of 12 cases would be required. Based on these considerations and the inclusion of an interim analysis, we set the target sample size at 12 subjects. We have revised the text in the “Sample size” section accordingly.

Comment 2: Outcomes: The definition of the primary endpoint is too generic and not consistent with its statistical definition.

Response: We thank the reviewer for pointing out that the definition of the primary endpoint was too generic and not consistent with its statistical definition. We have revised the “Outcomes” section to provide a more precise definition of the primary endpoint as “the successful formation of an implantable Biotube, defined as retrieval of a structurally intact Biotube of sufficient length and quality judged suitable for use in bypass surgery by the investigators.”

Furthermore, we have revised the “Statistical Methods” section to clarify that the primary endpoint will be analyzed as the proportion of successful cases with a 95% confidence interval. Secondary and safety endpoints will be summarized descriptively. We believe that these revisions provide a clearer and statistically consistent definition of the endpoints and their planned analyses.

Comment 3: The protocol should clearly specify the statistical method planned for analyzing the primary endpoint. If no hypothesis testing is not planned for the exploratory study, it needs to state that clearly. Descriptive measures such as success rates can be reported.

Response: We thank the reviewer for pointing out the need to more clearly specify the statistical method for analyzing the primary endpoint. We agree with this suggestion. In the revised manuscript, we have clarified that the primary endpoint will be analyzed as the success rate, defined as the proportion of subjects from whom a Biotube suitable for implantation can be retrieved, and that this proportion will be presented together with a 95% confidence interval. We have also stated explicitly that, given the exploratory nature of this trial, no formal hypothesis testing is planned and that all analyses will be descriptive in nature. We believe that these revisions provide a clearer and statistically consistent description of the planned analyses.

Comment 4: The purpose of the interim analysis is also unclear. Is it intended for safety monitoring or a futility check? With only half of the planned sample size, the analysis may not have sufficient power to draw meaningful conclusions.

Response: We thank the reviewer for raising this important point regarding the purpose of the interim analysis. We agree that with a small number of subjects, the interim analysis cannot provide statistically powered conclusions. We have therefore clarified in the revised manuscript that the interim analysis is intended for safety monitoring and feasibility assessment only, and not for hypothesis testing or a futility check. At the time when 6 subjects have reached 12 weeks after Biotube implantation, we will descriptively summarize the Biotube-forming ability and short-term patency to ensure that no unexpected safety issues arise and that the procedure remains feasible. The results of the interim analysis will not be used for formal statistical inference but only to support the safe continuation of the study.

---

## [Editor Report · Decision Letter 2]

20 Oct 2025

Safety and efficacy of an iBTA-induced autologous Biotube® vascular graft and its preparation device BTM1 in below-the-knee bypass surgery for chronic limb threatening ischemia: A protocol for an open-label, single-arm, multicenter clinical trial

PONE-D-25-31042R2

Dear Dr. Shuto,

We’re pleased to inform you that your manuscript has been judged scientifically suitable for publication and will be formally accepted for publication once it meets all outstanding technical requirements.

Kind regards,

Yoshiaki Taniyama, MD, PhD

Academic Editor

PLOS ONE
---

## [Editor Report · Acceptance letter]

PONE-D-25-31042R2

PLOS ONE

Dear Dr. Shuto,

I'm pleased to inform you that your manuscript has been deemed suitable for publication in PLOS ONE. Congratulations! Your manuscript is now being handed over to our production team.

Kind regards,

on behalf of

Dr. Yoshiaki Taniyama

Academic Editor

PLOS ONE